# Transcriptional Targeting of Dendritic Cells Using an Optimized Human *Fascin1* Gene Promoter

**DOI:** 10.3390/ijms242316938

**Published:** 2023-11-29

**Authors:** Yanira Zeyn, Dominika Hobernik, Ulrich Wilk, Jana Pöhmerer, Christoph Hieber, Carolina Medina-Montano, Nadine Röhrig, Caroline F. Strähle, Andrea K. Thoma-Kress, Ernst Wagner, Matthias Bros, Simone Berger

**Affiliations:** 1Department of Dermatology, University Medical Center of the Johannes Gutenberg University (JGU) Mainz, 55131 Mainz, Germany; yanira.zeyn@uni-mainz.de (Y.Z.); domihobernik@web.de (D.H.); chieber@uni-mainz.de (C.H.); gmedinam@students.uni-mainz.de (C.M.-M.); n.roehrig@uni-mainz.de (N.R.); 2Pharmaceutical Biotechnology, Department of Pharmacy, Center for NanoScience, Ludwig-Maximilians-Universität (LMU) Munich, 81377 Munich, Germany; ulrich.wilk@cup.uni-muenchen.de (U.W.); jana.poehmerer@cup.uni-muenchen.de (J.P.); ernst.wagner@cup.uni-muenchen.de (E.W.); 3Institute of Clinical and Molecular Virology, Friedrich-Alexander-Universität (FAU) Erlangen-Nürnberg, 91054 Erlangen, Germany; caroline.mohr@freenet.de (C.F.S.); andrea.thoma-kress@uk-erlangen.de (A.K.T.-K.)

**Keywords:** *fascin1*, cytomegalovirus, promotor activity, plasmid DNA, gene delivery, polyplex, polyethylenimine, dendritic cell, tumor cell

## Abstract

Deeper knowledge about the role of the tumor microenvironment (TME) in cancer development and progression has resulted in new strategies such as gene-based cancer immunotherapy. Whereas some approaches focus on the expression of tumoricidal genes within the TME, DNA-based vaccines are intended to be expressed in antigen-presenting cells (e.g., dendritic cells, DCs) in secondary lymphoid organs, which in turn induce anti-tumor T cell responses. Besides effective delivery systems and the requirement of appropriate adjuvants, DNA vaccines themselves need to be optimized regarding efficacy and selectivity. In this work, the concept of DC-focused transcriptional targeting was tested by applying a plasmid encoding for the luciferase reporter gene under the control of a derivative of the human *fascin1* gene promoter (pFscnLuc), comprising the proximal core promoter fused to the normally more distantly located DC enhancer region. DC-focused activity of this reporter construct was confirmed in cell culture in comparison to a standard reporter vector encoding for luciferase under the control of the strong ubiquitously active cytomegalovirus promoter and enhancer (pCMVLuc). Both plasmids were also compared upon intravenous administration in mice. The organ- and cell type-specific expression profile of pFscnLuc versus pCMVLuc demonstrated favorable activity especially in the spleen as a central immune organ and within the spleen in DCs.

## 1. Introduction

Based on a deeper understanding of the pathophysiology and the important role of immune dysfunction in cancer development and progression, nucleic acid-based immunotherapeutic approaches have attracted growing interest in recent years [1,2,3,4,5,6,7,8,9,10,11,12,13,14,15]. Hereby, the aims are (i) to modulate the immune system in such a way that it effectively fights the tumor and (ii) to create an immunological memory for long-term protection. DNA-based vaccination of dendritic cells (DCs) and other antigen-presenting cells (APCs) with transgenes that encode tumor-associated or -specific antigens and nucleic acid-based or encoded adjuvants aims to induce adaptive anti-tumor T cell immune responses [1]. So far, the overall efficacy of such DNA vaccines in clinical trials has been limited [1,16,17]. Their combination with other immunotherapies (e.g., check-point inhibitors, chimeric antigen receptor (CAR) T cells, or oncolytic viruses) or with conventional cancer treatments (e.g., chemotherapy, irradiation) may improve the clinical outcome [1]. Moreover, DNA vaccines can be optimized by (i) the careful choice of suitable antigens, (ii) the selection of appropriate adjuvants to stimulate APCs [18,19,20,21,22,23,24], (iii) the use of efficient delivery systems (e.g., nanoparticles like polyplexes [25,26,27,28,29,30]), and (iv) the structural optimization of the DNA itself. The latter can be achieved by using a strong APC-specific promoter that confines tumor antigen expression to APC. By this, unwanted gene expression in tumor-induced immune-regulatory cell types (e.g., tumor-associated macrophages or myeloid-derived suppressor cells), which in turn can induce tumor immune tolerance, may be prevented [1]. As DCs are the most potent T cell stimulatory APC population [31], they represent an ideal target for such an approach. Previously, the promoter of the murine *fascin1* (*Fscn1*) gene was identified to be suitable for transcriptional targeting of activated DCs [32]. Fscn1, a cytoskeletal actin-bundling protein, is evolutionarily highly conserved, expressed upon activation of DCs, and is relevant for their motility, migration, and T cell stimulatory activity, as recently reviewed [33]. Fscn1 is also required by tumor cells for metastasis [34]. Therefore, Fscn1 inhibitors have been developed and clinically tested for tumor therapy [35]. Interestingly, the pharmacological inhibition of Fscn1 activity in tumor-bearing mice has been reported to promote anti-tumor activity also due to direct effects on intratumoral DCs [36]. Wang and co-worker demonstrated an inhibitor-dependent accumulation of DCs within the tumor that internalized more tumor antigen and upon co-treatment with the PD-1 checkpoint inhibition stimulated cytotoxic T lymphocytes (CTLs).

In the current work, we aimed to assess the suitability of the human *Fscn1* gene promoter [37] to achieve transcriptional targeting of *Fscn1*-expressing cell types with regard to the translational perspective of DNA-based vaccination in the future. The concept of DC-focused transcriptional targeting was tested in vitro and in vivo by using a plasmid encoding for the firefly luciferase reporter gene under the control of an optimized derivative of the human *Fscn1* gene promoter (pFscnLuc), in comparison to a standard firefly luciferase encoding plasmid under the control of the strong ubiquitously active cytomegalovirus promoter and enhancer (pCMVLuc). As effective non-viral plasmid DNA (pDNA) carriers in vitro and in vivo, linear polyethylenimine 22 kDa (LPEI) [38,39] as well as succinylated branched polyethylenimine 25 kDa (succPEI, 10% succinylation degree) [40] were used. Both cationic polymers are able to efficiently compact pDNA into nanoparticulate complexes (so-called polyplexes [25]) via electrostatic interactions. Altogether, our study demonstrated preferable DC-focused activity of pFscnLuc both in vitro an in vivo compared to the standard CMV promoter.

## 2. Results and Discussion

### 2.1. Firefly Luciferase Reporter Gene Construct under the Control of an Optimized Human Fscn1 Gene Promoter

We have previously reported that both the murine [32,41] and human [37] *Fscn1* gene promoters allowed for transcriptional targeting of DCs. Subsequent studies demonstrated that plasmids containing the murine *Fscn1* promoter to drive antigen expression favorably stimulated a Th1-biased immune response [41,42,43]. With regard to subsequent in vivo studies in mice, we first assessed the activity of human *Fscn1* gene promoter reporter constructs in the murine DC-like cell line DC2.4 (Figure 1A). DC2.4 cells are known to be of mature phenotype [32,44] and thus are considered to express *Fscn1*, which was confirmed by us via intracellular Fscn1 staining (Figure 2A and Appendix A). We have previously shown that a 1.6 kb fragment of the human *Fscn1* gene promoter (phF1.6) (Figure 1A) evoked stronger activity in human DCs than the 3.1 kb encompassing parental vector [37]. In murine DC2.4 cells, phF1.6 showed cross-species activity and exerted similar activity as the positive control (pGL3-Ctrl) (Figure 1B, left). Further, we have reported that DC-specific activity was mediated by an enhancer region located within a 170 bp sequence stretch at the 5’-end of phF1.6 (Figure 1A). When fusing this enhancer sequence with the core promoter of the human *Fscn1* gene promoter as apparent in phF0.21, the resulting construct pFscnLuc (previously described as phF-TRR in [45]) exerted stronger transcriptional activity in DC2.4 cells than the parental constructs (2.5-fold higher compared to phF1.6; 5-fold higher compared to phF0.21). This enhancer/core promoter hybrid retained DC-focused transcriptional activity as observed in *Fscn1*-negative cell lines (i.e., macrophage-like P388 cells (Figure 1B, middle) and B cell-like A20 cells (Figure 1B, right)), yielding only about double activity as the negative control (pGL3-Basic) in either case.

In the following, the optimized human *Fscn1* gene promoter was evaluated in comparison to the standard CMV promoter in terms of its transcriptional activity in vitro and in vivo.

### 2.2. Tumor Cell Lines Express Fscn1 at Lower Levels Than DCs

Under homoeostatic conditions, *Fscn1* expression is mainly restricted to neuronal/glia cells and activated DCs, mediating membrane protrusions and the formation of filopodial extensions [46,47,48]. As mentioned above, the de novo expression of *Fscn1* is also often found in tumor cells and correlates with cancer aggressiveness by promoting cancer progression, migration, and metastasis [34,46].

To prove that the murine neuroblastoma cell line N2a and the murine hepatoma cell line Hepa1-6 express *Fscn1* and hence are suitable for pFscnLuc transfections, intracellular Fscn1 detection was conducted on cytospun cells. CLSM imaging revealed that both tumor cell lines indeed express *Fscn1* at a certain yet lower extent compared to DC2.4 cells (Figure 2A and Appendix A), and thus can be used for further evaluation of the human *Fscn1* promoter.

### 2.3. PEI Derivatives Are Suitable Carriers for pDNA Delivery

For the comparison of the optimized human *Fscn1* promoter with the standard CMV promoter in vitro and in vivo, LPEI [38,39,49] as the established gold standard for pDNA transfections as well as succPEI [40] were selected as pDNA carriers. Both cationic polymers were able to compact either plasmid at in vitro and in vivo concentrations into well-defined, homogenous nanoparticles by electrostatic interactions with hydrodynamic diameters (z-average) between 55 and 140 nm, polydispersity indices (PdIs) below 0.3, and a positive surface charge (zeta-potential ranging from +10 to +30 mV) (Appendix A).

Despite the potent transfection capability of PEI, its multi-faceted cytotoxicity can be an issue [49]. Furthermore, PEI-mediated effects on the immune system such as complement activation [50] or interactions with Toll-like receptors (TLRs) [51,52,53] have been reported, mediating either immune toxicity or immune stimulation [49]. Lowering the N/P (nitrogen-to-phosphate) ratio of LPEI pDNA complexes from highly effective N/P 9–10 [38,54,55,56] to N/P 6 [39] may be a compromise between efficacy and toxicity. Another option to reduce toxicity could be the surface modification of PEI [49] as for example demonstrated by the succinylation of branched PEI (succPEI) [40].

In the following, LPEI and succPEI were tested regarding their possible toxicity. The viability of DC-like DC2.4 cells treated overnight with LPEI and succPEI complexed with pFscnLuc at different N/P (LPEI; N/P 6 and 9, resp.) or *w*/*w* (weight-to-weight) ratios (succPEI; *w*/*w* 1.5 and 4, resp.) was moderately (~35%) reduced for LPEI but not for succPEI compared to untreated control cells (Appendix A). In agreement, DC2.4 cells showed reduced metabolic activity in response to overnight incubation with either polyplex formulation in a plasmid-independent manner, particularly at higher carrier amounts (LPEI N/P 9, succPEI *w*/*w* 4) (Appendix A). In contrast, we observed no carrier- or plasmid-dependent effects on the metabolic activity of the two *Fscn1*-expressing tumor cell lines (Appendix A).

### 2.4. The Optimized Human Fscn1 Promoter Confers Enhanced Reporter Gene Expression in Fscn1-Expressing Cell Lines

The cell-type-specific activity of pFscnLuc in comparison to standard plasmid pCMVLuc [57] was evaluated in the different *Fscn1*-expressing cell lines (see Figure 2A and Appendix A). In parallel assays, LPEI [49,54] as well as succPEI [40] were used as potent transfection agents at the indicated N/P (LPEI; N/P 6 and 9, resp.) or *w*/*w* ratios (succPEI; *w*/*w* 1.5 and 4, resp.). As depicted in Figure 2B, the DC-focused activity of pFscnLuc was confirmed for all four formulations since corresponding polyplexes mediated up to 7.4-fold higher RLU values in DC2.4 cells than polyplexes formed with the standard plasmid pCMVLuc. In contrast to DC2.4 cells, the preference for the *Fscn1* promoter plasmid was less pronounced in the neuroblastoma cell line N2a (Figure 2C) and in the hepatoma cell line Hepa1-6 (Figure 2D). This is in line with the lower *Fscn1* expression level compared to DC2.4 cells (Figure 2A and Appendix A). Yet, pFscnLuc was expressed in both tumor cell lines at almost similar luciferase expression levels as the standard plasmid pCMVLuc (Figure 2C,D). Altogether, these findings suggest on the one hand that the optimized *Fscn1* promoter was suitable to drive stronger transgene expression in DCs as mediated by the standard CMV promoter/enhancer, allowing for DC-focused DNA vaccination. On the other hand, the observation that the *Fscn1* promoter was also highly active in tumor cells may pave the way to employ DNA therapeutics that (co-)target tumor cells. For example, transgenes under the control of the *Fscn1* promoter encoding for anti-tumor cytokines (e.g., tumor necrosis factor (TNF)-α) can be used to transcriptionally address tumor cells, leading to apoptosis [1,58,59,60,61]. Hence, one could envision a DNA vaccine which encodes for a tumor antigen and at the same time for TNF-α, driving the activation of transfected DCs and at the same time exerting tumoricidal effects in the case of transfected tumor cells.

### 2.5. The Optimized Human Fscn1 Promoter Mediates Preferential Reporter Expression in DCs also In Vivo

#### 2.5.1. LPEI at Higher N/P Ratio Exerts Moderate Toxicity and Activation of DCs at the Same Time

In light of the potential cytotoxicity of LPEI polyplexes on DC2.4 cells, we tested for the corresponding effects of LPEI and succPEI also on bone marrow-derived (BM) DCs. Besides viability, also the expression of activation markers comprising MHCII (major histocompatibility complex) for antigen-presentation and CD80 as well as CD86 as costimulatory receptors for antigen-specific T cell activation [62,63,64] was analyzed. Despite the far higher carrier amounts (*w*/*w* ratios of 1.5 and 4 represent N/P ratios of ~11.25 and ~30), succPEI polyplexes were well tolerated by BMDCs, whereas LPEI especially at an N/P ratio of 9 moderately reduced their viability by around 30% (Appendix A). Nevertheless, in contrast to succPEI, LPEI also enhanced the expression of CD86 by means of the viable BMDC (at N/P 9, 1.5-fold increase compared to untreated control cells) (Appendix A), whereas the MHCII and CD80 expression levels remained largely unaffected (Appendix A). This effect may be due to an intrinsic activating potential of the LPEI polyplexes. In this regard, it has been reported that PEI stimulated DCs via TLR5 [52] and macrophages via TLR4 [51,53]. Further, it cannot be ruled out that necrotic cells within the culture released endogenous danger factors like HMGB1 (high mobility group box-1 protein), which binds to TLR4 as well [65], thereby triggering DC activation.

Due to these findings, the next isolated murine primary splenic immune cells were evaluated in a pre-experiment for subsequent in vivo studies. Splenic immune cell types were distinguished based on the expression of the corresponding lineage surface markers. In accordance with moderate cytotoxicity of LPEI polyplexes at N/P 9 in DC2.4 cells (Appendix A) and BMDCs (Appendix A), we observed a decrease in the viability of primary DCs by around 30% as compared to untreated control cells (Figure 3A). On the contrary, LPEI polyplexes at lower N/P ratio and succPEI polyplexes in general had no significant effect on DC viability. Opposed to DCs, the viability of other splenic immune cell types was not significantly affected by either polyplex as assessed for macrophages (MAC), polymorphonuclear neutrophils (PMN), B cells and natural killer cells (NK) (Figure 3A).

In accordance with our observations on BMDCs, LPEI polyplexes at an N/P ratio of 9 conferred a significant increase (~1.5-fold compared to untreated control cells) in CD86 activation marker expression on splenic DCs (Figure 3B). Since only DCs were activated, it is more likely that LPEI at N/P 9 exerted intrinsic stimulatory activity since a release of danger signals by necrotic cells may have broadly activated most of the monitored cell types.

#### 2.5.2. The Optimized Human Fscn1 Gene Promoter Evokes Reporter Gene Expression Preferably in Spleens

The performance of the *Fscn1* and the CMV promoter was compared also in vivo by the intravenous administration of corresponding LPEI and succPEI pDNA polyplexes in BALB/c mice. Despite showing moderate toxicity in DCs (see Section 2.3 and Section 2.5.1), LPEI polyplexes at an N/P ratio of 9 were considered as suitable for subsequent in vivo evaluation also with regard to their presumably intrinsic immunostimulatory potential. It is known from the literature that LPEI at N/P ratios of 9–10 is highly active in vivo and mediates only minor, if any, toxic effects [38,55,56]. For reasons of comparability and based on preliminary in vivo results (unpublished data), a *w*/*w* ratio of 1.5 was chosen for succPEI polyplexes, which corresponds to an N/P ratio of ~11.25 for unsubstituted PEI. The two carriers promoted different luciferase expression profiles (Figure 4A, Appendix A) with lower overall luciferase expression levels for the succPEI group (esp. in the lung; 210-fold for CMV, 22-fold for *Fscn1*). LPEI showed a strong preference for confer luciferase reporter expression in the lungs. In the LPEI group, the CMV promoter was ubiquitously active at a high level. In contrast, the *Fscn1* promoter mediated an overall lower luciferase expression, especially in the lungs and liver (10-fold less RLU), but conferred luciferase expression in the spleen as high as that observed for the CMV promoter. This indicates a clear shift towards expression in the spleen for the *Fscn1* promoter construct, directing expression away from the lung towards the spleen, which constitutes a suitable target organ containing high numbers of APCs. In the case of succPEI polyplexes, the *Fscn1* promoter favored reporter gene expression (12.6-fold higher RLU compared to CMV promoter) in the spleen. RLU levels in the lungs and liver remained constant, suggesting a preferable shift towards spleen expression mediated by the *Fscn1* promoter. Spleen/lung and spleen/liver luciferase expression ratios underline the spleen preference in both the LPEI and the succPEI group (Appendix A).

#### 2.5.3. The Optimized Human *Fscn1* Gene Promoter Shows DC-Focused Activity In Vivo

In addition, promoter-dependent reporter expression activity was examined in single-cell suspensions retrieved from the spleen, lymph nodes, liver, and lungs of accordingly treated BALB/c mice via intracellular immunostaining with a luciferase-specific antibody (Figure 4B and Appendix A). succPEI was used as carrier for this evaluation, since it showed a more selective spleen activity compared to LPEI in the ex vivo luciferase assay (Figure 4A, Appendix A). Moreover, we observed that succPEI-complexed pFscnLuc conferred a significantly stronger splenic luciferase expression than succPEI/pCMVLuc transfection complexes (12.6-fold increase, *p* value = 0.0102; Appendix A). pFscnLuc expression was determined in DCs alongside macrophages, both representing APC-like cell lines, and detected in DC populations of all tested organs (Appendix A). Standard plasmid pCMVLuc was also highly expressed due to the strong and ubiquitously active CMV promoter, not in favor of a real benefit of pFscnLuc at first glance. However, a comparison of calculated ratios of luciferase expressing DC/MAC in the single organs revealed a slightly better performance of pFscnLuc with preferential luciferase expression in DCs, particularly in the spleen and lymph nodes (Figure 4B). For example, the DC:MAC ratio in the spleen was approx. 2.7 for pFscnLuc compared to approx. 1.9 for pCMVLuc.

## 3. Materials and Methods

### 3.1. Materials

#### 3.1.1. Plasmids

Plasmids pCMVLuc [57] (encoding *Photinus pyralis* firefly luciferase under control of cytomegalovirus promotor and enhancer) and pFscnLuc (described as phF-TRR in [45]; encoding *Photinus pyralis* firefly luciferase under the control of a derivative of the human *Fscn1* gene promotor) were obtained from Plasmid Factory GmbH (Bielefeld, Germany). Promoter-less expression vector pGL3-Basic (promoter-less expression vector encoding for *Photinus pyralis* firefly luciferase), and pGL3-Control (encoding *Photinus pyralis* firefly luciferase under the control of the hybrid SV40 promoter/enhancer) (all from Promega, Mannheim, Germany) were obtained from Plasmid Factory GmbH (Bielefeld, Germany) as endotoxin-free preparations. The *Renilla reniformis* luciferase encoding vector pRL-EF1α [37] was amplified using the Endofree Plasmid Maxi Kit (Qiagen, Hilden, Germany).

#### 3.1.2. Chemicals

HEPES (4-(2-hydroxyethyl)-1-piperazineethanesulfonic acid) was obtained from Biomol (Hamburg, Germany), disodium ethylenediaminetetraacetic acid (EDTA) as well as glucose from Merck (Darmstadt, Germany). Cell culture consumables were obtained from Faust Lab Science (Klettgau, Germany) or Greiner Bio-One (Frickenhausen, Germany). Cell culture media, bovine serum albumin (BSA), fetal bovine serum (FBS), antibiotics, and trypsin/EDTA were purchased from Sigma Aldrich (Munich, Germany) and PAN-Biotech (Aidenbach, Germany); β-mercaptoethanol was obtained from Roth (Karlsruhe, Germany). Cell culture 5× lysis buffer, CellTiter-Glo Reagent, and D-luciferin sodium salt were obtained from Promega (Mannheim, Germany); dithiothreitol (DTT), adenosine 5′-triphosphate (ATP) disodium salt trihydrate, coenzyme A trilithium salt, and protease and phosphatase inhibitor cocktail were from Sigma-Aldrich (Munich, Germany). In the case of co-transfection using firefly luciferase and renilla luciferase encoding vectors, luciferase activities were determined using the Dual-luciferase reporter assay kit (Promega, Mannheim, Germany).

#### 3.1.3. Transfection Agents

Succinylated branched PEI 25 kDa (succPEI; succinylation degree of 10%) and unmodified linear PEI 22 kDa (LPEI) were synthesized and analyzed as described previously [40,66]. The starting products branched PEI (brPEI) 25 kDa as well as poly(2-ethyl-2-oxazoline) were purchased from Sigma Aldrich (Munich, Germany). The transfection reagent JetPEI (linear PEI) was obtained from Polyplus (Illkirch, France).

#### 3.1.4. Cell Lines

The murine neuroblastoma cell line Neuro2a (N2a; Cellosaurus database entry: https://www.cellosaurus.org/CVCL_0470; accessed on 30 October 2023), the murine macrophage-like cell line P388 (Cellosaurus database entry: https://www.cellosaurus.org/CVCL_7222; accessed on 30 October 2023), and the murine B cell-like cell line A20 (Cellosaurus database entry: https://www.cellosaurus.org/CVCL_1940; accessed on 30 October 2023) were purchased from the American Type Culture Collection (ATCC; Manassas, VA, USA). The murine hepatoma cell line Hepa1-6 (Cellosaurus database entry: https://www.cellosaurus.org/CVCL_0327; accessed on 30 October 2023) was obtained from ATCC (Gaithersburg, MD, USA), and the murine DC-like cell line DC2.4 (Cellosaurus database entry: https://www.cellosaurus.org/CVCL_J409; accessed on 30 October 2023) was from Merck Millipore (Darmstadt, Germany).

#### 3.1.5. Antibodies

PE-/APC- and PE-Cy7-labeled CD11c-specific (clone N418), FITC-CD86 (GL-1), SB600-CD11b (M1/70), PE-NK1.1 (PK136), PE-eFl610-Ly6G (1A8-L6g), V500-B220 (RA3-6B2), SB702-CD45 (30-F11), eFl506- or FITC-CD3 (17-A2), AF647-Luciferase (EPR17789), eFl450-F4/80 (BM8), AF488-CD19 (6D5), AF647-XCR1 (ZET), PE-Cy7-CD172a (P84), and PE-CD32b (AT130-2) antibodies were used for flow cytometric analysis (FACS). FACS antibodies were purchased from BD Biosciences (Franklin Lakes, NJ, USA), BioLegend (San Diego, CA, USA), or ThermoFisher (Waltham, MA, USA). eFl780-FVD (fixable viability dye) used to identify dead cells was obtained from ThermoFisher. Unlabeled mouse anti-human Fscn1 antibody (55K2; Sigma-Aldrich, Munich, Germany), a corresponding isotype control antibody (mouse IgG1, clone MOPC-21, BioLegend), and secondary AF488-labeled IgG goat anti-mouse antibody (948492; ThermoFisher) were used for confocal laser scanning analysis (CLSM).

### 3.2. pDNA Polyplex Formation

pDNA and calculated amounts of LPEI at indicated N/P (nitrogen/phosphate) ratios or of succPEI at indicated *w*/*w* (weight/weight) ratios, respectively, were diluted with HBG (HEPES buffer with glucose; 20 mM of HEPES, 5% (*w*/*v*) glucose; pH 7.4) in separate vials at equal volumes. pDNA and LPEI or succPEI solutions were mixed by rapid pipetting and incubated for 30 min at room temperature (RT) in a closed vial.

*Note:* In the case of succPEI, the N/P ratio was considered to be not suitable for calculation of the amounts of succPEI to pDNA since some amino groups are substituted with succinic acid and thus not involved in nucleic acid binding. Therefore, the *w/w* ratio was used instead of the commonly used N/P ratio [40]. A *w/w* ratio of 1 corresponds to an N/P ratio of ~7.5 for unsubstituted PEI.

### 3.3. Physico-Chemical Characterization of pDNA Polyplexes

Polyplexes were formulated in HBG buffer as described above (see Section 3.2.) at pCMVLuc concentrations used for the in vitro experiments (i.e., 10 µg mL^−1^ in the case of transfections in tumor cell lines N2a and Hepa1-6; 25 µg mL^−1^ in the case of transfections in the DC-like cell line DC2.4) and for the in vivo studies (i.e., 300 µg mL^−1^). Measurements of size and zeta-potential were performed by dynamic and electrophoretic laser-light scattering (DLS, ELS) using a Zetasizer Nano ZS (Malvern Instruments, Malvern, Worcestershire, UK) in a folded capillary cell (DTS1070). Size and polydispersity index were measured in 100 µL polyplex solution using the following instrument settings: equilibration time 30 s, temperature 25 °C, refractive index 1.330, and viscosity 0.8872 mPa·s. Samples were measured three times with six sub runs per measurement. For measurement of the zeta-potential, all samples were diluted to 800 µL with HBG directly before measurement. Parameters were identical to the size measurement apart from an equilibration time of 60 s. Three measurements with 15 sub runs lasting 10 s each were performed, and zeta-potentials were calculated with the Smoluchowski equation.

### 3.4. Cell Culture

Immortalized cell lines. N2a and Hepa1-6 cells were grown in Dulbecco’s Modified Eagle’s Medium (DMEM)-low glucose (1 g L^−1^ glucose) supplemented with 10% (*v*/*v*) FBS, 4 mM of stable glutamine, 100 U mL^−1^ of penicillin, and 100 µg mL^−1^ of streptomycin. The DC2.4 cell line was grown in Iscove’s Modified Dulbecco’s Medium (IMDM) supplemented with 10% (*v*/*v*) FBS, 4 mM of stable L-glutamine, 100 U mL^−1^ of penicillin, 100 µg mL^−1^ of streptomycin, and 50 µM of β-mercaptoethanol. Bone marrow-derived cells (BMDCs, 2 × 10^5^ mL^−1^) were seeded in 12-well plates (Greiner Bio-One, Frickenhausen, Germany) in an IMDM-based culture medium (see above) including recombinant murine granulocyte-macrophage colony-stimulating factor (GM-CSF; 10 ng mL^−1^; Miltenyi Biotec, Bergisch Gladbach, Germany) to differentiate DCs. DC culture media were replenished on days three and six of culture. Cells were cultured at 37 °C and either 5% (cell lines) or 10% (BMDC) CO_2_ in an incubator with a relative humidity of 95%.

### 3.5. Confocal Laser Scanning Microscopy (CLSM)

Cytospins were generated, and *Fscn1* expression was detected by CLSM, as described previously [46]. In brief, cells were re-suspended (5 × 10^5^ mL^−1^) in PBS (phosphate-buffered saline; 136.9 mM NaCl, 2.7 mM KCl, 8.1 mM Na_2_HPO_4_, and 1.5 mM KH_2_PO_4_). Sets of 50,000 cells were cytospun (500 rpm, 5 min, RT) onto microscope slides (Superfrost Plus; VWR, Darmstadt, Germany) using a Cytospin 3 (ThermoFisher, Waltham, MA, USA), air-dried overnight, and stored at −20 °C. For staining, cytospins were incubated first with pre-cooled methanol (Roth, Karlsruhe, Germany) for 10 min to permeabilize the cell membrane and washed twice with PBS. During all subsequent incubation steps, samples were kept in a humidified chamber. Cytospins were incubated with a Fc receptor blocking antibody (clone 2.4-G2; 1:50-diluted in PBS/2% (*v*/*v*) BSA) for 10 min at RT. Then, an Fscn1-specific or isotype control antibody (each 1:50-diluted in PBS/2% (*v*/*v*) FBS) was added, and samples were incubated for 20 min at RT. Afterwards, samples were washed and incubated with AF488-labeled secondary anti-mouse antibody (1:400-diluted in PBS/2% (*v*/*v*) FBS) for 20 min at RT. After two washing steps with PBS, HCS Cell Mask Orange (2 µg mL^−1^, ThermoFisher, Waltham, MA, USA) was applied and samples were incubated for 30 min and finally washed with water. As a control, samples were left untreated or incubated with one agent only, respectively. All samples were covered with fluorescence mounting medium (DAKO; Agilent, Santa Clara, CA, USA) and a coverslip. For CLSM, samples were imaged using a Leica SP8 confocal microscope (Mannheim, Germany) equipped with a 20/0.75 NA air objective using a 405 nm laser for transmission images, a 488 nm laser for AF488 (Fscn1) excitation (emission and detection within a spectral window of 499–581 nm), and 552 nm laser exposure for AF555 (HCS Cell Mask Orange) excitation (emission and detection within a spectral window of 562–632 nm). Cellular Fscn1 intensities were calculated using Imaris software version 9.3.1 (Bitplane, Zurich, Switzerland).

### 3.6. In Vitro Transfection and Luciferase Detection

One day prior to transfection, 10,000 N2a, 10,000 Hepa1-6, or 5000 DC2.4 cells were seeded per well into 96-well plates. For transfection, 200 ng (N2a, Hepa1-6) or 500 ng (DC2.4) pDNA per well were applied, respectively. HBG served as a negative control. All transfection experiments were performed in biological triplicates. The medium was replaced with 80 µL of fresh medium containing 10% (*v*/*v*) FBS, and polyplexes formed at indicated N/P (LPEI) or *w*/*w* (succPEI) ratios in 20 µL HBG, as described above (see Section 3.2.), were added to each well. The plates were incubated at 37 °C and 5% CO_2_ without a change in medium. After 24 h of incubation, the medium was removed, cells were lysed with 100 µL of cell culture 0.5× lysis buffer (Promega, Mannheim, Germany), and the samples were frozen at −80 °C at least overnight. Prior to measurement of the luciferase activity, the plates were equilibrated for 1 h at RT. Luciferase activity in 35 µL of the cell lysate was measured for 10 s by using a Centro LB 960 plate reader luminometer (Berthold Technologies, Bad Wildbad, Germany) after the addition of a 100 µL LAR buffer (20 mM glycylglycine; 1 mM MgCl_2_; 0.1 mM EDTA; 3.3 mM DTT; 0.55 mM ATP; 0.27 mM coenzyme A, pH 8–8.5), supplemented with 5% (*v*/*v*) of a mixture of 10 mM luciferin and 29 mM glycylglycine. Transfection efficiency was calculated for the seeded number of cells and presented as relative light units (RLU) per well.

In some experiments (results displayed in Figure 1), cells (DC2.4, P388, A20; each 20,000 per well in a 24-well plate) were co-transfected with firefly (950 ng of largest vector and equimolar amounts of smaller vector) and renilla luciferase (50 ng) encoding vectors using JetPEI (Polyplus, Illkirch, France), as recommended by the manufacturer. On the next day, plates were centrifuged, media were removed, and 100 µL of Passive Lysis Buffer (Promega, Mannheim, Germany) was applied to each well. In total, 10 µL of a cell lysate were assayed for luciferase activities by subsequently applying firefly and renilla substrate as recommended by the manufacturer (Promega, Mannheim, Germany) and measuring luciferase activities every 5 s. Firefly luciferase activities were divided by renilla luciferase activities to account for differences in transfection efficacy.

### 3.7. Metabolic Activity of Transfectd Cells

Cell transfections were performed as described in Section 3.6. The supernatant was removed at 24 h after transfection, and 25 µL of medium as well as 25 µL of CellTiter-Glo Reagent were added to each well. After incubation on an orbital shaker for 30 min at RT, luminescence was recorded using a Centro LB 960 plate reader luminometer (Berthold Technologies, Bad Wildbad, Germany). The luminescent signals (in RLU) of the samples were set in relation to the luminescent signal of the negative control (HBG buffer-treated control cells). Results are presented as relative metabolic activity related to the negative control. Experiments were performed in biological triplicate.

### 3.8. In Vivo Appliction of pDNA Polyplexes

In vivo experiments were performed according to the guidelines of the German Animal Welfare Act and were approved by the animal experiments ethical committee of the Government of Upper Bavaria (accreditation number Gz. ROB-55.2-2532.Vet_02-19-19). Six-week-old female BALB/c mice (Janvier Labs, Le Genest-Saint-Isle, France) were randomly divided into groups of five. The mice were housed in isolated ventilated cages under specific pathogen-free conditions with a 12 h day/night interval, and food and water ad libitum. Weight and general well-being were monitored continuously. The experiments were performed by intravenous tail vein injection of polyplexes formed at indicated N/P (LPEI) or *w*/*w* (succPEI) ratios, as described above (see Section 3.2), respectively, each containing 60 µg of pCMVLuc or pFscnLuc in 200 µL HBG. Mice were euthanized at 24 h after injection. The organs were dissected and washed carefully with PBS, followed by analysis via an ex vivo luciferase gene expression assay of the lungs, liver, and spleen (see Section 3.9) as well as an immunostaining of single cell suspensions retrieved from dissected organs (lungs, liver, spleen, and lymph nodes; see Section 3.10 and Section 3.11).

### 3.9. Ex vivo Luciferase Gene Expression Assay after In Vivo Transfection

For the detection of luciferase activity, organs were homogenized in Luciferase Cell Culture Lysis Reagent 1× (Promega, Mannheim, Germany), supplemented with 1% (*v*/*v*) protease and phosphatase inhibitor cocktail (Sigma-Aldrich, Munich, Germany) using a tissue and cell homogenizer (FastPrep^®^-24, MP Biomedicals, Santa Ana, CA, USA). Then, the samples were frozen overnight at −80 °C to ensure full lysis. The samples were thawed and centrifuged for 10 min at maximum speed (~13,000 rpm) at 4 °C. Luciferase activity in 25 µL supernatant was measured for 10 s, as described above (see Section 3.6). The luciferase expression is presented as RLU per gram (g) of organ after background subtraction (lysis buffer).

### 3.10. Singe Cell Preparation after In Vivo Transfection

Spleens and lymph nodes were meshed using a pestle and a 40 µM cell strainer (Greiner Bio-One, Frickenhausen, Germany) to yield a single-cell suspension. Spleen cells (2 × 10^6^ in 500 µL) were kept in FACS tubes overnight in a medium (IMDM, containing 5% (*v*/*v*) FBS, 2 mM of L-glutamine, 100 U mL^−1^ of penicillin, 100 µg mL^−1^ of streptomycin, and 50 µM ß-mercaptoethanol). Murine liver non-parenchymal cells (NPCs) were enriched by liver dissociation, as described previously [67]. In brief, an enzymatic dissociation mixture (Liver Dissociation Kit; Miltenyi Biotec, Bergisch-Gladbach, Germany) was pre-incubated for 15 min in a water bath at 37 °C. Then, the liver tissue was dissected in little pieces in the dissociation mixture, subsequently transferred into prepared C tubes (Miltenyi Biotec). The C tubes were placed into a gentleMACS Dissociator to be minced (program m_liver_03; Miltenyi Biotec). The derived cell suspension was incubated under continuous shaking for 30 min at 37 °C, followed by another round of gentleMACS-mediated dissociation (m_liver_04). After erythrocyte lysis, liver NPCs were enriched by density centrifugation employing 30% Histodenz-HBSS (both from Sigma-Aldrich, Munich, Germany). Lung cell suspensions were obtained using the Lung Dissociation Kit (Miltenyi Biotec) according to the manufacturer’s protocol.

### 3.11. Flow Cytometry

To assess living cells via flow cytometry, cells were washed in staining buffer (PBS, 1% (*v*/*v*) FBS, 0.5 mM EDTA) and were incubated with Fc receptor blocking antibody (2.4-G2) for 15 min at RT. Then, samples were incubated with fluorescence-labeled antibodies for 20 min at 4 °C. Afterwards, samples were washed with PBS and subsequently incubated with eFl780-FVD to identify dead cells. For intracellular staining, samples were fixed with 0.5 mL Fluorofix Fixation Buffer for 20 min and washed twice with 0.5 mL Perm Wash Buffer (both from BioLegend, San Diego, CA, USA). Then, the antibody for intracellular luciferase detection was added, and the samples were incubated for 20 min at RT in the dark. Finally, samples were washed two times with every 1 mL of Perm Wash Buffer. Gating was performed according to the strategies described in Appendix A. Samples were assayed using an Attune NxT flow cytometer, and data were analyzed using Attune NxT software v3.2.1 (both ThermoFisher, Waltham, MA, USA).

### 3.12. Statistics

Results are presented as arithmetic mean and standard deviation (SD) or standard error of the mean (SEM), respectively, of at least three experiments. Unpaired Student’s two-tailed *t*-test with Welch’s correction, as well as ordinary one-way ANOVA (multiple comparison, Tukey test), respectively, were performed using GraphPad Prism^TM^ 9.5.1 in order to analyze statistical significances between groups. Significance levels are indicated with symbols: ns, not significant; *p* > 0.05; * or # *p* ≤ 0.05; ** *p* ≤ 0.01; *** *p* ≤ 0.001; **** *p* ≤ 0.0001.

## 4. Conclusions

The efficacy of DNA vaccines in clinical trials is still low [1,16,17] but may be improved by transcriptional targeting of DCs using a DC-specific promoter. In the current study, we evaluated a hybrid enhancer/core human *Fscn1* gene promoter in this regard. This derivative showed preferable DC-focused activity in vitro compared to the standard CMV promoter. Upon systemic administration in mice, the expression profile of pFscnLuc was encouraging, with favorable activity in the spleen as a central immune organ, and in DCs at a somewhat higher level than in macrophages. This suggests that the human *Fscn1* gene promoter may be suitable for DC-focused transcriptional targeting. Moreover, pFscnLuc mediated high reporter gene expression in *Fscn1*-expressing tumor cells. Further, we observed carrier-dependent differences in the immunostimulatory potential, toxicity, as well as the in vivo luciferase expression profile, which are to be analyzed in more detail in follow-up studies.

Dual transcriptional targeting of DNA to both DCs and tumor cells may be an option to enhance the efficiency of DNA vaccines. This could be realized by employing DNA vaccines that encode for a tumor antigen to induce anti-tumor T cell responses in transfected DCs [1]. In transfected tumor cells, this may enhance antigen presentation via MHCI, thereby supporting recognition by tumor antigen-specific CTLs. Further, a DNA vaccine needs to co-deliver an adjuvant to ensure activation of the APC to prevent antigen-specific tolerance induction [68]. In the case of a DNA vaccine that may co-transfect APC and tumor cells, this adjuvant could be, for example, a TNF-α encoding gene, promoting DC activation by engaging TNF receptor 2 [69,70]. In tumor cells, TNF-α may induce apoptosis due to TNF receptor 1 signaling [60,61]. Alternatively, the vaccine may encompass the sequence for a short hairpin (sh)RNA that inhibits inhibitory signaling, e.g., of the transcription factor signal transducer and activator of transcription (STAT3). STAT3 knockdown in DCs has been reported to promote their anti-tumor activity [71]. Further, in tumor cells, STAT3 inhibition counteracted tumor growth and tumor resistance towards various therapeutic approaches [72]. Besides the design of nanovaccines that may transcriptionally co-address DCs and tumor cells, either cell type may be specifically targeted by conjugation of the nanocarrier system with corresponding targeting moieties [73,74,75]. In this regard, the attachment of antibodies that bind, e.g., CD205 [76,77,78], or of mannose or trimannose units, which address mannose recognizing receptors CD206 (macrophage mannose receptor) and CD209 (DC-SIGN, dendritic cell-specific intercellular adhesion molecule-3-grabbing non-integrin) [79,80,81,82,83,84,85,86], has been shown by us and others to yield the DC-focused internalization of a nanoformulation. In a complementary manner, we and others demonstrated preferential targeting of tumor cells by nanoformulations, e.g., by means of the conjugation of folate [75].

As a perspective, the specificity and efficacy of nanotherapeutics may be considerably improved by using receptor-addressing nanoformulations that deliver transcriptional targeting pDNA. This approach would aim to combine both levels of cell targeting to circumvent the side effects resulting from biological activity of the cargo in unwanted cell types.

## Figures and Tables

**Figure 1 ijms-24-16938-f001:**
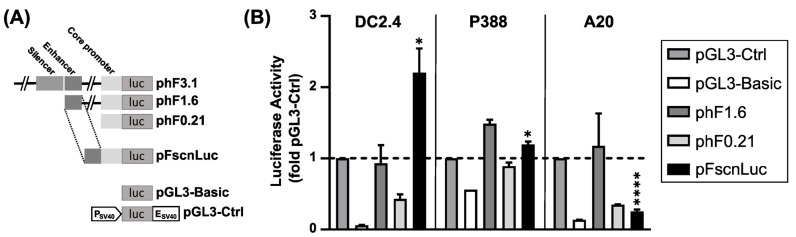
Activity of human *Fscn1* promoter reporter constructs. (**A**) Scheme of human *Fscn1* gene promoter luciferase reporter constructs (phF) derived from parental phF3.1 in comparison to control vectors pGL3-Basic (promoter-less) and pGL3-Ctrl (hybrid SV40 promoter/enhancer). (**B**) DC-like DC2.4 cells, macrophage-like P388 cells, and B cell-like A20 cells (all of murine origin; each 20,000 cells/well) were transfected with *Fscn1* promoter firefly luciferase reporter constructs as indicated and control vectors (pGL3-Basic, pGL3-Ctrl) at equimolar amounts normalized to 950 ng of the largest plasmid, respectively, and a constant amount of 50 ng of a renilla luciferase encoding control vector (pRL-EF1α [37]) using the transfection reagent JetPEI according to the manufacturer’s protocol (Polyplus, Illkirch, France). On the next day, firefly luciferase activities were determined. Data denote the relative luciferase activity (fold pGL3-Ctrl; *n* = 4; mean + SEM). Significant differences versus pGL3-Ctrl: * *p* ≤ 0.05; **** *p* ≤ 0.0001 (unpaired Student’s two-tailed *t*-test with Welch’s correction; GraphPad Prism^TM^ 9.5.1).

**Figure 2 ijms-24-16938-f002:**
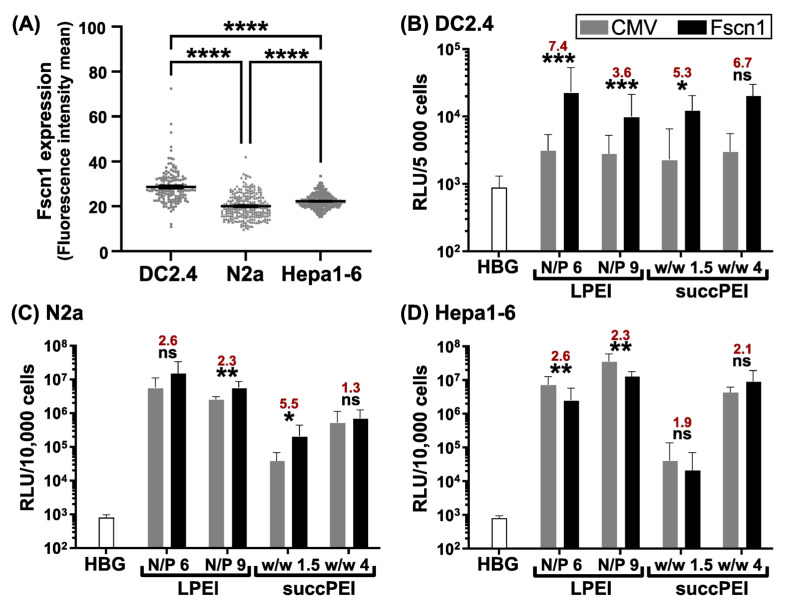
Quantification of Fscn1 levels and promoter-dependent reporter activities in DC-like (DC2.4) and tumor (N2a, Hepa1-6) cell lines (all of murine origin). (**A**) Data denote the corresponding fluorescence intensities per cell and the mean ± SEM of 154–301 cells per group after CLSM imaging. Corresponding cytospin images are displayed in Appendix A. (**B**–**D**) Luciferase gene expression (*n* = 3; mean + SD) was assayed in comparison to HBG buffer-treated control cells at 24 h after transfection with LPEI (N/P 6 and 9, resp.) and succPEI (*w*/*w* 1.5 and 4, resp.) polyplexes containing pFscnLuc or pCMVLuc, resp., in the DC-like cell line DC2.4 (**B**) at a pDNA concentration of 500 ng/well, as well as in the neuroblastoma cell line N2a (**C**) and the hepatoma cell line Hepa1-6 (**D**) at a pDNA concentration of 200 ng/well. Red numbers indicate the fold difference in RLU values between pFscnLuc and pCMVLuc. *Note*: For succPEI, *w*/*w* ratios of 1.5 and 4 represent N/P ratios of ~11.25 and ~30 of an unsubstituted PEI. (**A**–**D**) Significance levels are indicated as follows: (**A**) **** *p* ≤ 0.0001 (one-way ANOVA, Tukey test, GraphPad Prism^TM^ 9.5.1); (**B**–**D**) significant differences between both plasmids: * *p* ≤ 0.05; ** *p* ≤ 0.01; *** *p* ≤ 0.001; ns, not significant (unpaired Student’s two-tailed *t*-test with Welch’s correction; GraphPad Prism^TM^ 9.5.1).

**Figure 3 ijms-24-16938-f003:**
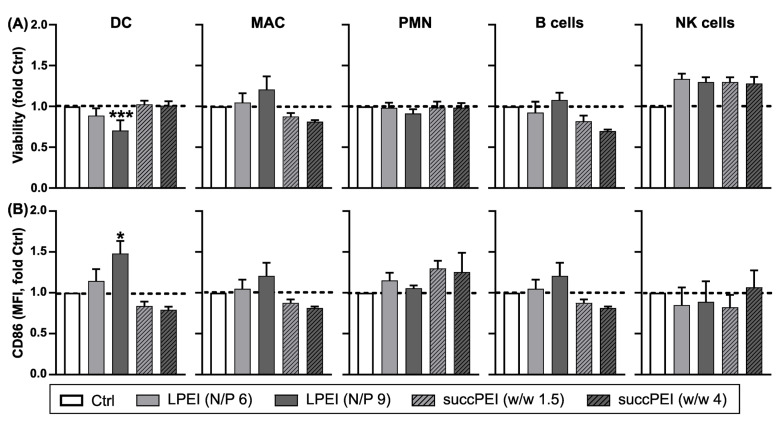
Effect of LPEI and succPEI polyplexes on the viability and activation of splenic immune cells. Splenic immune cells (2 × 10^6^ cells per well) were incubated with LPEI and succPEI polyplexes at the indicated ratios using 1 µg pFscnLuc/well. On the next day, the viability (**A**) and CD86 expression (**B**) of leukocyte populations was assessed by flow cytometry. Splenic immune cell populations were delineated by sequential gating, as described in Appendix A. Graphs denote (**A**) the frequencies of FVD (fixable viability dye)-negative viable cells normalized to the value of untreated cells (Ctrl) (*n* = 4; mean + SEM) and (**B**) the MFI (mean fluorescence intensity) of CD86 expression in relation to the according expression level of untreated cells (Ctrl) (each *n* = 4; mean + SEM). (**A**,**B**) Significant differences versus Ctrl: * *p* ≤ 0.05; *** *p* ≤ 0.001 (one-way ANOVA, Tukey test; GraphPad Prism^TM^ 9.5.1). DC, dendritic cells; MAC, macrophages, PMN, polymorphonuclear cells. NK, natural killer cells. *Note:* For succPEI, *w*/*w* ratios of 1.5 and 4 represent N/P ratios of ~11.25 and ~30 of an unsubstituted PEI.

**Figure 4 ijms-24-16938-f004:**
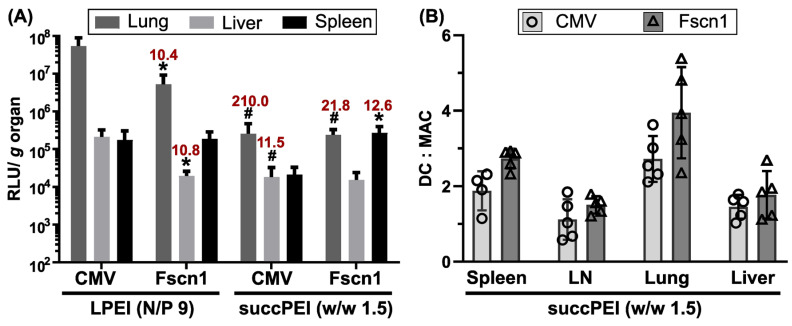
In vivo comparison of *Fscn1* and CMV promoter activities in BALB/c mice. (**A**) Luciferase expression in organs (*n* = 5; mean + SD) assessed via an ex vivo luciferase assay at 24 h after intravenous injection of LPEI (N/P 9) and succPEI (*w*/*w* 1.5) polyplexes, resp., formed at a pDNA dose of 60 µg/animal in 200 µL of HBG buffer. Significant differences between pCMVLuc and pFscnLuc: within the LPEI or succPEI group, resp.: * *p* ≤ 0.05 (unpaired Student’s two-tailed *t*-test with Welch’s correction, GraphPad Prism^TM^ 9.5.1); significant differences between LPEI and succPEI for CMV or *Fscn1* promoter, resp.: # *p* ≤ 0.05 (unpaired Student’s two-tailed *t*-test with Welch’s correction, GraphPad Prism^TM^ 9.5.1). Red numbers indicate the fold difference in RLU values between the compared groups of the significance tests. For clarity, all significance levels and fold changes are displayed in Appendix A. (**B**) Comparison of *Fscn1* and CMV promoter activities on the single cell level. Ex vivo analysis of single cell suspensions retrieved from different organs at 24 h after the intravenous injection of 200 µL of succPEI polyplexes (*w*/*w* 1.5), containing 60 µg pDNA. Cells were isolated and subjected to flow cytometric analyses. Gating strategies for the cell suspensions retrieved from the distinct organs are pictured in Appendix A. Ratios of luciferase-expressing DCs (dendritic cells) to MACs (macrophages) of *n* = 5 are displayed. The ratio calculation is based on the data shown in Appendix A. LN, lymph nodes.

## Data Availability

The data presented in this study are available from the corresponding authors on request.

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
