# Peer review of "Transcriptional Targeting of Dendritic Cells Using an Optimized Human Fascin1 Gene Promoter"

_ijms, 2023, doi:10.3390/ijms242316938_

Round 1
Reviewer 1 Report
Comments and Suggestions for Authors
In this manuscript Zeyn et al, analyzed different constructs based on the fascin promoter and different carriers for pDNA delivery for their cell targeting properties using luciferase reporter as a surrogate of transcription efficiency. The authors found a suitable combination of construct and carriers targeting dendritic cells and tumor cells. The results are technically sound and well-presented but, in my opinion the scope of the results, fall short. The rationale for such design should be better explained. DC are characterized by high expression of fascin and therefore it is not surprising that the construct using fascin promoter shows higher expression in this cell type. Which type of gene should be included in a DNA vaccine based on this approach? How would the authors be able to target DC without targeting tumor cells or vice versa? Probably side effects would occur.
Recent works (PMID: 37509352 and 33826900) dealing with the role of fascin in DC should have been considered in this manuscript to discuss about the cell targeting of construct carrying fascin promoter.
Author Response
In this manuscript Zeyn et al, analyzed different constructs based on the fascin promoter and different carriers for pDNA delivery for their cell targeting properties using luciferase reporter as a surrogate of transcription efficiency. The authors found a suitable combination of construct and carriers targeting dendritic cells and tumor cells. The results are technically sound and well-presented but, in my opinion the scope of the results, fall short. The rationale for such design should be better explained. DC are characterized by high expression of fascin and therefore it is not surprising that the construct using fascin promoter shows higher expression in this cell type. Which type of gene should be included in a DNA vaccine based on this approach? How would the authors be able to target DC without targeting tumor cells or vice versa? Probably side effects would occur.
> We thank the reviewer for this hint. We have extended the Conclusion section discussing first the outcome of dual transcriptional targeting of DC and tumor cells with nano-carriers that co-deliver tumor antigen and adjuvant, giving as examples TNF-a and STAT3 shRNA that may exert beneficial effects on DC activation and at the same time inhibit tumor cell viability via different mechanisms (lines 341-352). Then we discuss that nano-carriers may be conjugated with targeting moieties to address DC (lines 353-357) or tumor cells (lines 357-358) giving according examples. We consider that nano-carriers designed to address distinct cell types via appropriate targeting units plus transcriptional targeting of the according cell type may improve overall specificity and efficacy of nanoformulations (lines 359-363).
Recent works (PMID: 37509352 and 33826900) dealing with the role of fascin in DC should have been considered in this manuscript to discuss about the cell targeting of construct carrying fascin promoter.
> We cite now both references in the Introduction section by mentioning that pharmacological Fscn1 inhibition bas been demonstrated to exert on the one hand direct inhibitory effects on the tumor and oin the other hand to boost anti-tumor immune responses by promoting the T cell stimulatory activity of DC.
Reviewer 2 Report
Comments and Suggestions for Authors
Zeyn and colleagues conducted research in which they utilized a plasmid encoding the luciferase reporter gene controlled by the regulatory element of human fascin1 (referred to as pFscnLuc). Their objective was to enhance the specificity and effectiveness of DNA vaccines targeting the transcription in dendritic cells (DCs). When comparing pFscnLuc to the widely active cytomegalovirus promoter and enhancer (pCMVLuc) in both cell lines and mice, the authors found that pFscnLuc demonstrated more favorable activity, particularly in the spleen DCs. While this discovery is intriguing and the data presentation is clear, some revisions are necessary.
The major comments are as follows:
1. In the results section, the subtitles should be rewritten as conclusions rather than descriptions of the methodology.
2. On line 255-259, the authors said ‘In the case of succPEI polyplexes, the Fscn1 promoter favored reporter gene expression (12-fold higher RLU compared to CMV promoter) in the spleen as a suitable target organ containing high numbers of APCs. In the case of LPEI polyplexes, pFscnLuc exerted significantly lower lung and liver expression than pCMVLuc (10-fold less RLU).’
In addition to preference of organ, LPEI and succPEI also affected the performance of the two promoters (CMV and Fscn1 did not show differences in spleen using LEPI. But Fscn1 outperformed with succPEI in spleen. In liver, LEPI led to a better performance of CMV and succPEI did not result in any differences between CMV and Fscn1). The differences in organ preference and promoter performance related to polyplex choice require clarification.
The minor comments are as follows:
1. In Figure S1, higher-resolution images are needed, and concurrent phase-contrast images should be provided to visualize the cells.
2. In Figure S3, the authors should determine whether the decreases in MHCII, CD80, and CD86 expression induced by succPEI are statistically significant.
3. On lines 208-211, the authors suggest possible reasons for the observed effects. To address these questions, they should conduct experiments with LPEI without plasmid to compare CD86 expression and assess HMGB1 and TLR4 expression levels in both the culture medium and cells.
4. In Figure 4 and Figure S4, the authors compared luciferase signal in mouse organs and cells derived from these organs. To enhance the specificity of the promoter, they should also display the pFscnLuc expression levels in other cell types alongside DC and macrophage cells in the single-cell assay. Additionally, they should clarify the comparisons indicated by the red numbers made in Figure 4A using a bar or a table.
Author Response
Zeyn and colleagues conducted research in which they utilized a plasmid encoding the luciferase reporter gene controlled by the regulatory element of human fascin1 (referred to as pFscnLuc). Their objective was to enhance the specificity and effectiveness of DNA vaccines targeting the transcription in dendritic cells (DCs). When comparing pFscnLuc to the widely active cytomegalovirus promoter and enhancer (pCMVLuc) in both cell lines and mice, the authors found that pFscnLuc demonstrated more favorable activity, particularly in the spleen DCs. While this discovery is intriguing and the data presentation is clear, some revisions are necessary.
The major comments are as follows:
- In the results section, the subtitles should be rewritten as conclusions rather than descriptions of the methodology.
> The subtitles are revised as suggested.
- On line 255-259, the authors said ‘In the case of succPEI polyplexes, the Fscn1 promoter favored reporter gene expression (12-fold higher RLU compared to CMV promoter) in the spleen as a suitable target organ containing high numbers of APCs. In the case of LPEI polyplexes, pFscnLuc exerted significantly lower lung and liver expression than pCMVLuc (10-fold less RLU).’ In addition to preference of organ, LPEI and succPEI also affected the performance of the two promoters (CMV and Fscn1 did not show differences in spleen using LEPI. But Fscn1 outperformed with succPEI in spleen. In liver, LEPI led to a better performance of CMV and succPEI did not result in any differences between CMV and Fscn1). The differences in organ preference and promoter performance related to polyplex choice require clarification.
> We re-wrote the paragraph for better understandability. Spleen preference is further underlined by fold-changes (new Table S2) and LUC expression ratios (new Table S3).
“[…] The two carriers promoted different luciferase expression profiles (Figure 4A, Table S2) with lower overall luciferase expression levels for the succPEI group (esp. in the lung; 210-fold for CMV, 22-fold for Fscn1). LPEI showed a strong preference to confer luciferase reporter expression in the lungs. In the LPEI group, the CMV promoter was ubiquitously active at high level. In contrast, the Fscn1 promoter mediated overall lower luciferase expression, especially in lung and liver (10-fold less RLU), but conferred luciferase expression in the spleen as high as observed for the CMV promoter. This indicates a clear shift towards expression in the spleen for the Fscn1 promoter construct, directing expression away from the lung towards the spleen, which constitutes a suitable target organ containing high numbers of APCs. In the case of succPEI polyplexes, the Fscn1 promoter favored reporter gene expression (12.6-fold higher RLU compared to CMV promoter) in the spleen. RLU levels in the lung and liver remained constant, suggesting a preferable shift towards spleen expression mediated by pFscnLuc. “Spleen/lung” and “spleen/liver” luciferase expression ratios underline the spleen preference in both the LPEI and the succPEI group (Table S3).”
Table S2. Significance levels & fold-changes of the data displayed in Fig. 4.
|
Significance levels |
Fold-change |
||||
|
p value |
|||||
|
LPEI
CMV vs Fscn1 |
Lung |
0.0377 |
* |
10.4 |
(-) |
|
Liver |
0.0174 |
* |
10.8 |
(-) |
|
|
Spleen |
0.8585 |
ns |
1.1 |
(+) |
|
|
succPEI
CMV vs Fscn1 |
Lung |
0.8632 |
ns |
1.1 |
(-) |
|
Liver |
0.6982 |
ns |
1.2 |
(-) |
|
|
Spleen |
0.0102 |
* |
12.6 |
(+) |
|
|
CMV
LPEI vs succPEI |
Lung |
0.0284 |
* |
210.0 |
(-) |
|
Liver |
0.0166 |
* |
11.5 |
(-) |
|
|
Spleen |
0.0584 |
ns |
8.1 |
(-) |
|
|
Fscn1
LPEI vs succPEI |
Lung |
0.0500 |
* |
21.8 |
(-) |
|
Liver |
0.4268 |
ns |
1.3 |
(-) |
|
|
Spleen |
0.2708 |
ns |
1.4 |
(+) |
|
(-), fold-decrease; (+), fold-increase.
Table S3. Spleen/lung and spleen/liver luciferase expression ratios.
|
LUC expression ratios *10 |
|||
|
spleen/lung |
spleen/liver |
||
|
LPEI |
CMV |
0.03 |
8.25 |
|
Fscn1 |
0.36 |
96.03 |
|
|
succPEI |
CMV |
0.84 |
11.68 |
|
Fscn1 |
11.42 |
176.40 |
|
The minor comments are as follows:
- In Figure S1, higher-resolution images are needed, and concurrent phase-contrast images should be provided to visualize the cells.
> We revised Fig. S1 as suggested.
- In Figure S3, the authors should determine whether the decreases in MHCII, CD80, and CD86 expression induced by succPEI are statistically significant.
> There are no statistical differences for succPEI compared to control and thus not indicated.
- On lines 208-211, the authors suggest possible reasons for the observed effects. To address these questions, they should conduct experiments with LPEI without plasmid to compare CD86 expression and assess HMGB1 and TLR4 expression levels in both the culture medium and cells.
> We consider testing LPEI alone without plasmid as not really relevant as it is applied in DNA complexes. But we agree that it would be interesting to confirm our hypothesis with the DNA complexes. Yet, this is beyond scope of the current manuscript, as in the current study the focus was clearly on the evaluation of transcriptional targeting by using an optimized DNA vaccine construct. However, this question will be addressed in follow-up studies as we also plan to analyze the other observed carrier-dependent differences in more detail.
- In Figure 4 and Figure S4, the authors compared luciferase signal in mouse organs and cells derived from these organs. To enhance the specificity of the promoter, they should also display the pFscnLuc expression levels in other cell types alongside DC and macrophage cells in the single-cell assay. Additionally, they should clarify the comparisons indicated by the red numbers made in Figure 4A using a bar or a table.
> Encouraged by the results of the ex vivo luciferase assay which revealed spleen preference, we wanted to prove that LUC is indeed expressed in dendritic cells. Thus, we evaluated LUC expression on the single cell level in dendritic cells in comparison to macrophages as second APCs in a first proof-of-concept study. We didn’t test for expression in other cell types but will address this in follow-up studies.
> We clarified the comparisons in Fig. 4A as suggested by providing now significance levels and fold-changes between the compared groups in a tableà see Table S2.